# Multiple Pathways in the Control of the Shade Avoidance Response

**DOI:** 10.3390/plants7040102

**Published:** 2018-11-17

**Authors:** Giovanna Sessa, Monica Carabelli, Marco Possenti, Giorgio Morelli, Ida Ruberti

**Affiliations:** 1Institute of Molecular Biology and Pathology, National Research Council, 00185 Rome, Italy; giovanna.sessa@uniroma1.it (G.S.); monica.carabelli@uniroma1.it (M.C.); 2Research Centre for Genomics and Bioinformatics, Council for Agricultural Research and Economics (CREA), 00178 Rome, Italy; marco.possenti@crea.gov.it (M.P.); giorgio.morelli.crea@gmail.com (G.M.)

**Keywords:** Arabidopsis, auxin, HD-Zip transcription factors, light environment, photoreceptors

## Abstract

To detect the presence of neighboring vegetation, shade-avoiding plants have evolved the ability to perceive and integrate multiple signals. Among them, changes in light quality and quantity are central to elicit and regulate the shade avoidance response. Here, we describe recent progresses in the comprehension of the signaling mechanisms underlying the shade avoidance response, focusing on Arabidopsis, because most of our knowledge derives from studies conducted on this model plant. Shade avoidance is an adaptive response that results in phenotypes with a high relative fitness in individual plants growing within dense vegetation. However, it affects the growth, development, and yield of crops, and the design of new strategies aimed at attenuating shade avoidance at defined developmental stages and/or in specific organs in high-density crop plantings is a major challenge for the future. For this reason, in this review, we also report on recent advances in the molecular description of the shade avoidance response in crops, such as maize and tomato, and discuss their similarities and differences with Arabidopsis.

## 1. Introduction

Plants, as sessile organisms, have evolved complex and sophisticated molecular processes to sense and react to the presence of neighboring plants. Plants can be divided into two groups depending on their response to competition for light: shade tolerance and shade avoidance [1,2,3]. To detect the presence of plants in close proximity, shade-avoiding plants use multiple cues [4]. Among these cues, changes in light intensity and quality play a central role in the regulation of the shade avoidance response. Light reflected or transmitted through photosynthetic plant tissues is depleted in blue (B), red (R), and UV-B wavelengths. Hence, the reflected or transmitted light is enriched in green (G) and far-red (FR) spectral regions, resulting in lowered ratios of R/FR light and B/G light. Plants perceive these differences through multiple photoreceptors, which in turn trigger signaling cascades to regulate plant growth under suboptimal light environments [5,6,7,8].

Arabidopsis is very responsive to FR-enriched light. At the early stage of seedling development, the perception of shade results in hypocotyl elongation, a reduction of cotyledon and leaf lamina expansion, and the diminution of root development (Figure 1). Here, we describe the key pathways underlying the shade avoidance response, focusing mainly on Arabidopsis, because most of the molecular processes regulating this response have been characterized in this model plant.

## 2. Photoreceptors in the Control of Shade Avoidance 

The R/FR ratio is a highly accurate indicator of plant proximity, and probably for this reason, for many years, shade avoidance research has mostly focused on the phytochrome signaling of changes in the R/FR ratio. However, a large number of evidence points to the reduced irradiance and the blue/green ratio as signals that play important roles in activating plant responses to canopy light [5,6,7,8].

### 2.1. Phytochromes

Phytochromes exist in two photo-convertible isoforms: a R light-absorbing form (Pr) and a FR light-absorbing form (Pfr). In the darkness, phytochromes are synthesized in the Pr form, which is inactive. After triggering with R light, the Pr form is converted into the active Pfr form, which, in turn, can absorb FR and switch back to Pr. The active Pfr form is translocated to the nucleus, giving rise to the responses [5,10].

The phytochrome apoproteins are encoded by a small gene family in the majority of plant species. In Arabidopsis, they are encoded by five genes, *PHYA*–*PHYE*. *PHYE* likely originated from a duplication within the *PHYB* lineage only in dicotyledonous plants. *PHYD*, which is closely related to *PHYB*, presumably emerged from a gene duplication within Brassicaceae [11]. *PHYC* probably arose from a duplication within the *PHYA* lineage [11]. phyA is rapidly degraded in its Pfr form, and signals during the conversion between the Pr and Pfr form mediated by the R/FR ratio light. phyB–E are all relatively stable in the Pfr form [5,10,12].

Among the light-stable phytochromes, phyB has a predominant role in the regulation of the shade avoidance response. However, evidence exists that phyD and phyE function redundantly with phyB in promoting shade-induced elongation [12,13] (Figure 2). By contrast, phyA attenuates the elongation response induced by low R/FR light [9,14,15,16] (Figure 2).

In the nucleus, phytochromes directly bind the Phytochrome-Interacting Factors (PIFs), which are a subfamily of basic Helix–Loop–Helix (bHLH) transcription factors involved in the control of plant growth and development [17,18,19]. The Arabidopsis genome encodes eight PIF/PIF-like proteins—PIF1, PIF3–8, and PIL1/PIF2—all containing a conserved active phytochrome B binding (APB) domain, which is required for the interaction with the Pfr form of phyB. PIF1 and PIF3 also contain an active phytochrome A binding (APA) domain, which is necessary and sufficient for binding the Pfr form of phyA. Most of the PIFs promote growth, whereas PIF6 and PIL1/PIF2 seem to have an opposite function [20]. PIF proteins have both redundant and distinct functions at different stages of plant development, and coherently, only a subset of target genes is regulated by multiple PIFs (PIF1, PIF3–5) [20]. PIFs bind to promoter regions enriched in the cis element G-box and the E-box variant, which is known as the PBE-box (PIF binding E-box) [18]. However, the mechanisms through which different PIF proteins specifically recognize distinct set of target genes are largely unknown. Interestingly, it has been recently shown that the promoters of PIF1 target genes are enriched with G-box coupling elements (GCEs), which bind PIF1-interacting transcription factors (PTFs). These interactions may contribute to the targeting of PIF1 to specific sites in the genome [21].

In most cases, the interaction of PIFs with phyB in the nucleus results in PIF’s phosphorylation and ubiquitination, leading to a fast degradation via the 26S proteasome [17]. PIF3, PIF4, and PIF5 protein levels increase rapidly in green seedlings upon inactivation of the phytochromes by simulated shade [22,23]. Instead, PIF7 is not rapidly degraded upon interaction with phyB in high R/FR light, but rather accumulates in a phosphorylated form. Exposure to low R/FR results in a rapid decrease of the amount of phosphorylated PIF7 with a concomitant increase in the level of dephosphorylated PIF7 [24]. PIF1, PIF3, PIF4, PIF5, and PIF7 have all been directly implicated in the shade avoidance response [22,23,24,25]. The shade-induced elongation response is indeed reduced in *pif4 pif5*, *pif1, pif3, pif4, pif5,* quadruple (*pifq*), and *pif7* loss-of-function mutants [22,23,24].

Interestingly, PIF proteins directly control the expression of both positive and negative regulators of the shade avoidance response [5,6,7,8,26] (Figure 2). 

Among the positive regulators is the *Homeodomain-Leucine Zipper* (*HD-Zip*) *Arabidopsis Thaliana HomeoBox2* (*ATHB2*) transcription factor gene, which is involved in the elongation response induced by light quality changes [27,28]. The *ATHB2* gene is rapidly and reversibly regulated by changes in the R/FR ratio light [29]. phyB, phyD, and phyE have all been implicated in the regulation of *ATHB2* by changes in the ratio of R/FR light [30]. *ATHB2* induction by FR-enriched light does not require de novo protein synthesis [31], and is significantly diminished in loss-of-function *pif* mutants (*pif4 pif5*; *pifq*) [22,32]. Furthermore, there is evidence that *ATHB2* is a direct target of PIF proteins [25]. Relevantly, among the positive regulators are also several auxin biosynthesis *YUCCA* (*YUC*) genes, thus directly linking the perception of shade light to plant growth [24].

Among the negative regulators of shade avoidance controlled by PIF proteins is Long Hypocotyl in Far Red 1/Slender In Canopy Shade 1 (HFR1/SICS1), which is an atypical bHLH protein. *HFR1/SICS1* is rapidly induced by FR-enriched light, and it has been demonstrated that it is recognized in vivo by PIF5 [25,33,34]. Prolonged exposure to Low R/FR leads to the accumulation of HFR1/SICS1 and the formation of non-active heterodimers with PIF4 and PIF5 [33,34]. Consistently, several genes that are rapidly and transiently induced by low R/FR are upregulated in loss-of-function *hfr1/sics1* mutants under persistent shade [33,35]. Moreover, *hfr1/sics1* plants display an exaggerated shade avoidance response, whereas transgenic seedlings overexpressing a stable HFR1/SICS1 protein have suppressed elongation [33,36]. Helix Loop Helix1/Phytochrome Rapidly Regulated1 (HLH1/PAR1) [31,33] is another atypical bHLH protein gene that also acts as a negative regulator of the shade avoidance response. It is rapidly upregulated by low R/FR light, without the requirement of de novo protein synthesis. HLH1/PAR1 has been proposed to act as an antagonist of bHLH transcription factors, including PIF4 [36,37,38,39].

The attenuation of shade avoidance responses also involves a low R/FR stimulation of phyA signaling [9,40,41] (Figure 2). The *PHYA* gene is early induced by low R/FR, and phyA is required for the upregulation of the basic leucine zipper (bZIP) transcription factor gene, Elongated Hypocotyl 5 (HY5), which is a central regulator of photomorphogenesis [42]. HY5, on one hand, downregulates genes induced early by low R/FR light, and on the other hand, positively regulates photomorphogenesis-promoting genes under persistent shade [9]. Evidence exists that HY5 binds to PIF proteins [43,44].

phyA in its active Pfr form directly interacts with Suppressor of PhyA-105 (SPA) proteins and inhibits their interaction with Constitutively Photomorphogenic 1 (COP1) [45]. The COP1/SPA complexes are part of the Cullin 4-Damaged DNA Binding 1 ubiquitin E3 ligase complex (CUL4–DDB1^COP1/SPA^), and are required for substrate recognition [46]. Several positive regulators of photomorphogenesis, including HY5 and HFR1/SICS1, are targeted for 26 proteasome-mediated degradation by CUL4–DDB1^COP1/SPA^ [41]. The active form of phyA also interacts with COP1 [45]. Evidence exist that the binding of COP1 and SPA proteins is relevant for the activity of CUL4–DDB1^COP1/SPA^. Therefore, it has been proposed that the direct interaction of phyA and SPA proteins inactivates CUL4–DDB1^COP1/SPA^, which in turn results in the stabilization of positive regulators of photomorphogenesis [41] (Figure 2). phyB in its active form has also been shown to bind to SPAs and inhibit their interaction with COP1 [45] (Figure 2). The analyses of loss-of-function *cop1* and *spa1-4* mutants in low R/FR indicate that the COP1/SPA complex is essential for shade-induced elongation [47,48]. It has been suggested that in low R/FR, reduced levels of the active form of phyB indirectly enhance PIF activity, increasing the COP1/SPA-mediated degradation of negative regulators of the shade avoidance response [48,49]. Together, the data indicate that the phyA and phyB-mediated control of COP1/SPA activity oppositely affect the levels of negative regulators of shade avoidance such as HY5, HFR1/SICS1, HLH1/PAR1, and members of the B-Box (BBX ) transcription factor family [50,51,52].

### 2.2. Cryptochromes

Cryptochromes are flavoprotein photoreceptors that were originally identified in Arabidopsis, and subsequently found in prokaryotes, archaea, and many eukaryotes [53]. Cryptochromes (CRY) are homologous to photolyases that catalyze light-dependent DNA repair [54]. The Arabidopsis genome encode two cryptochromes, CRY1 and CRY2. They consist of two domains, the PHR (photolyase-homologous region) domain, which is required for photoperception and dimer formation, and the CCE (cryptochrome C-terminal extension) domain, which is involved in signal transduction to downstream factors. It has been proposed that cryptochromes are activated by blue light through conformational changes, mostly in CCE domains [55]. Following blue light activation, CRY2 is rapidly degraded by the 26-proteasome system, whereas CRY1 is stable [54].

Both CRY1 and CRY2 are involved in low blue light (LBL)-induced shade avoidance response [56,57,58]. Interestingly, it has been recently demonstrated that PIF4 and PIF5 activity is required for LBL-induced hypocotyl growth, and evidence has been provided that these PIFs physically interact with CRY1 and CRY2 [58,59]. Furthermore, chromatin immunoprecipitation sequencing has shown that CRY2 binds to PIF4 and PIF5-regulated gene promoters [58]. Transcriptomic analysis revealed different expression profiles in low R/FR and LBL-treated seedlings. It is relevant that LBL, unlike low R/FR, does not involve changes in auxin levels and sensitivity, further supporting the proposal that phy and CRY photoreceptors control plant responses to shade via largely independent pathways [56,57,58].

Analogously to the active form of phyB, photoexcited CRY1 has been shown to bind to SPA1, resulting in the suppression of the SPA1–COP1 interaction. This in turn reduces COP1 activity, leading to increased levels of transcription factors such as HY5 [60].

### 2.3. UVR8

UV-B light is strongly filtered by plant canopies, thus providing further information on plant density [6,61]. In Arabidopsis, the inhibition of hypocotyl elongation by UV-B light depends on the UV-B receptor UVR8 [62,63]. UVR8 in its dimeric form perceives UV-B light; the absorption of UV-B induces the instant monomerization of the photoreceptor followed by interaction with COP1. This, in turn, promotes the accumulation of HY5 and its close relative HY5 Homologue (HYH) [64,65,66]. UVR8 promotes gibberellic acid (GA) degradation in a HY5/HYH-dependent manner, contributing to the stabilization of DELLA (where D is aspartic acid, E glutamic acid, L leucine, L leucine, A alanine) proteins and the consequent formation of inactive DELLA–PIF complexes [67]. Furthermore, evidence exist that UV-B also enhances the degradation of PIF4 and PIF5 [67]. Together, the data indicate that UV-B light inhibits PIF function, thereby attenuating plant responses to canopy shade [67,68].

## 3. HD-Zip Transcription Factors in the Control of Shade Avoidance 

The HD-Zip class of transcription factors appears to be present exclusively in the plant kingdom [69]. HD-Zip proteins form a dimeric complex that recognize pseudopalindromic DNA elements [70,71,72,73], and act as positive or negative regulators of gene expression [74]. The Arabidopsis HD-Zip proteins, on the basis of the sequence homology in the HD-Zip DNA-binding domain, the presence of other conserved motifs, and specific intron and exon positions, have been grouped into four families: HD-Zip I–IV [75,76,77,78,79,80]. The phylogenetic and bioinformatics analysis of HD-Zip genes using transcriptomic and genomic datasets from a large number of Viridiplantae species indicated that the HD-Zip class of proteins was already present in green algae [81].

All four HD-Zip protein families can be further classified into subfamilies consisting of paralogous genes that have likely originated through genome duplication, considering their association with chromosome-duplicated regions in Arabidopsis and rice [77,78,79,80]. Interestingly, members of both the HD-Zip II and HD-Zip III protein families have been implicated in the control of shade avoidance [74,82].

Relevantly, HD-Zip II and HD-Zip III binding sites share the same core sequence [70,76], thereby leading to the hypothesis that members of the two families may control the expression of common target genes [83]. HD-Zip II proteins contain an LxLxL (where L is leucine and x is another amino acid) type of Ethylene-responsive element binding factor-associated amphiphilic repression (EAR) motif [79,84], and there is strong evidence that they function as transcriptional repressors [27,83,85,86]. On the contrary, HD-Zip III transcription factors are considered activators of gene expression [73,83,87,88,89].

### 3.1. HD-Zips II

The HD-Zip II protein family includes *ATHB2*, which is the first gene shown to be rapidly and reversibly regulated by light quality changes [29]. phyB, phyD, and phyE are all involved in the regulation of *ATHB2* by low R/FR ratio light [29,30], and it has been shown that *ATHB2* is recognized in vivo by PIF5 [25]. A lack of ATHB2 function results in diminished hypocotyl elongation in low R/FR ratio light, whereas the phenotypes of seedlings with elevated levels of ATHB2 in high R/FR resembles that of wild type in shade [27,28]. The expression of ATHB2, as deduced by the β-glucuronidase (GUS) pattern observed in ATHB2:ATHB2:GUS seedlings, is rapidly and transiently induced by shade in all the cell layers of the hypocotyl [28]. This and other experimental evidence (see below) indicated that ATHB2 acts as a positive regulator of shade avoidance.

The HD-Zip II family consists of 10 genes, five of which [*ATHB2*, *Homeobox Arabidopsis Thaliana* (*HAT1*), *HAT2*, *ATHB4* and *HAT3*] are induced by low R/FR ratio light [79]. In the *hat3 athb4* double loss-of-function mutant hypocotyl elongation is impaired [90], whereas the overexpression of HAT1, HAT2, HAT3, and ATHB4 causes phenotypes that are analogous to those observed in plants with elevated levels of ATHB2 in high R/FR [26,35,79,86,90], further highlighting the redundancy of these proteins in the regulation of shade avoidance. Relevantly, homologue genes are induced in monocot and dicot plants by low R/FR ratio light, strongly suggesting that the function of HD-Zips II may be conserved through evolution [91,92,93].

Very recent work has shown that prolonged shade results in an early exit from proliferation in the first pairs of Arabidopsis leaves, and that this process depends on the action of ATHB2 and ATHB4 (Figure 3) [94]. 

Furthermore, evidence has been provided that ATHB2 and ATHB4 work in concert in the control of leaf development specifically in a low R/FR light environment, likely forming heterodimeric complexes as suggested by yeast two-hybrid assays [94,95]. The data provide novel insights on the molecular mechanisms underlying leaf development in shade. However, further work is needed to uncover the links between the ATHB2 and ATHB4 transcription factors and the known regulatory pathways involved in the control of leaf cell proliferation [96,97].

Links between HD-Zip II proteins and auxin have been established [35,74]. However, how HD-Zips II interact with auxin machineries is still largely unknown.

Interestingly, a growing body of evidence demonstrates that besides their function in plant growth responses to shade, HD-Zips II play a major role in key developmental processes in a sunlight simulated environment, including embryo apical development, shoot apical meristem (SAM) activity, organ polarity, and gynoecium development [74,83,98,99,100,101]. These studies suggest that developmental processes and shade avoidance responses, sharing these transcription factors, could be intertwined. Connections between developmental and shade avoidance regulatory networks are further indicated by the recent finding that under shade, PIFs directly suppress multiple *miR156* genes, resulting in the increased expression of the *Squamosa-Promoter Binding Protein-Like* (*SPL*) family of genes [102], which have a role in the regulation of several aspects of plant development [103]. 

### 3.2. HD-Zips III

The HD-Zip III protein family consists of five members: ATHB8, Corona (CNA), Phabulosa (PHB), Phavoluta (PHV), and Revoluta (REV). Several evidence have indicated that HD-Zip III proteins act as master regulators of embryonic apical fate [104], and are required to maintain SAM activity and establish lateral organ polarity [105,106]. The pattern of HD-Zips III expression largely overlaps with that of auxin distribution [89,107,108,109,110,111,112]. Furthermore, *HD-Zip III* genes are regulated at the post-transcriptional level by the microRNAs miR165/166, which negatively affect their expression through mRNA cleavage [105,113].

Interestingly, there is evidence that REV directly positively regulates *Tryptophan Aminotransferase of Arabidopsis 1* (*TAA1)* and *YUC5*, indicating that at least part of its role in plant development implies the regulation of auxin biosynthesis [73,114]. Relevantly, *TAA1* and *YUC5* are directly negatively regulated by KANADI1 (KAN1), which is a key determinant of abaxial cell fate in the leaf [57,115,116,117]. Furthermore, it has been recently demonstrated that genes implicated in auxin transport, including the influx carriers *LIKE Auxin Resistant 2* (*LAX2*) and *LAX3*, and response are also direct targets of REV [89,112,114,115].

Among the genes directly regulated by REV are also *HAT3*, *ATHB4*, *ATHB2*, and *HAT2*, and there is evidence that PHB and PHV are involved in the regulation of *HAT3* [73,83]. Coherently, the HAT3 and ATHB4 expression pattern in simulated sunlight essentially coincides with that of PHB, PHV, and REV. *ATHB2* expression is instead restricted to procambial cells early during embryo and leaf development; however, *ATHB2* is expressed in the *HAT3* and *ATHB4* domains in the *hat3 athb4* mutant, compensating in part for the lack of HAT3 and ATHB4 [83]. 

The direct regulation of *HD-Zip II* genes by HD-Zip III transcription factors and the finding that the phenotypes of *hat3 athb4 athb2* loss-of-function *HD-Zip II* mutants in sunlight resemble those of *rev phb phv* indicate that HD-Zip II and HD-Zip III proteins function in the same pathways under a sun-simulated environment [74,82]. Considering that HD-Zip II proteins work as negative regulators of gene expression [27,83,85], it was proposed that they may restrict HD-Zip III expression [74]. Interestingly, it was recently shown that REV, which is expressed exclusively in the adaxial side of the leaf because of the activity of microRNA (miR) 165/166 in the abaxial leaf domain, physically interacts with HAT3 and ATHB4 to directly repress the expression of *MIR165/166* genes in the adaxial side [118]. 

The analysis of *HD-Zip III* loss-of-function and gain-of function mutants has uncovered the involvement of REV in shade-induced elongation growth. *rev* loss-of-function mutants as well as plants ectopically expressing *MIR165a* display reduced elongation growth under simulated shade, whereas REV gain-of-function mutants (*rev10D*) show slightly long hypocotyl phenotypes under simulated sunlight [73,82]. It will be of interest in the future to investigate whether HD-Zip II and HD-Zip III proteins act together in the regulation of gene expression under a simulated shade environment.

## 4. Auxin as a Driver of the Shade Avoidance Response

There is a large body of evidence showing that plant responses to shade involve changes in hormonal pathways. Here, we focus on auxin, whereas for other hormones involved in the shade avoidance response, we recommend recent reviews [119,120]. Auxin has a central role in many responses induced by neighbor detection and canopy shade, such as the increased elongation of hypocotyl and petioles, and reduced leaf and root growth. Auxin homeostasis, transport, and signaling are all regulated in response to shade [35,121]. Interestingly, it has been shown that whereas the increase in auxin synthesis is a major event at the early stages of shade avoidance, the persistence of shade mainly results in the modulation of auxin sensitivity [25,122,123,124].

### 4.1. Auxin Homeostasis

Exposure to shade results in a rapid increase in the levels of auxin [24,25,125]. New auxin is synthesized in cotyledons from tryptophan (Trp) through TAA1, which is an enzyme encoded by the *Shade Avoidance3* (*SAV3*) gene [125,126]. Trp is converted to indole-3-pyruvic acid (IPA), and IPA in turn is modified to indole-3-acetic acid (IAA) by the action of the YUC family of flavin monooxygenases [127,128,129,130]. *YUC2*, *YUC5*, *YUC8*, and *YUC9* are rapidly regulated by low R/FR ratio light through PIF transcription factors [24,125]. Furthermore, the *sav3* mutant and the quadruple *yuc2 yuc3 yuc8 yuc 9* mutant are impaired in low R/FR-induced responses [125,131,132].

Low R/FR ratio light also controls auxin homeostasis by modulating its inactivation. Indeed, a number of auxin-inducible genes of the Gretchen Hagen 3 (GH3) family are quickly upregulated by low R/FR [14,133]. GH3 proteins promote the reduction of the free IAA pool by the conjugation of IAA to different amino acids [134], and it has been reported that GH3 mutants show defects in the elongation responses of the hypocotyl to light [135,136]. Furthermore, it has been recently shown that the loss-of-function of *VAS2* [*IAA-amido synthetase* (*GH3.17*)] results in an increase in free IAA at the expense of IAA-glutamate in the hypocotyl epidermis. Interestingly, the *vas2* mutants display longer hypocotyls in response to low R/FR light largely independently of the novo IAA biosynthesis in cotyledons, demonstrating the relevance of local auxin metabolism to modulate IAA homeostasis in an organ-specific manner in response to shade [137].

The relevance of local responses is also demonstrated by the recent finding that the alteration of the R/FR ratio at the leaf tip induces an upwards leaf movement that is confined to the leaf perceiving the light signal. Evidence have been provided that this hyponastic response depends on the synthesis of auxin in the leaf and its transport to the petiole [138,139].

### 4.2. Auxin Transport

It has been proposed that auxin that is synthesized in the cotyledons through the TAA1/YUC pathway upon low R/FR exposure is transported to hypocotyls, where it stimulates cell elongation [125]. Consistent with this proposal, auxin transport inhibitors abolish low R/FR-induced elongation, highlighting the relevance of auxin distribution for shade avoidance [27,125].

A large body of evidence indicates that the active transport of auxin is strictly controlled during neighbor detection and canopy shade. A low R/FR light ratio regulates the expression of the polar-auxin-transport efflux carriers PIN-Formed (PIN) 1, PIN3, PIN4, and PIN7 [14,25,133,140,141]. Moreover, the triple loss-of-function *pin3 pin4 pin7* mutant does not elongate under simulated shade [131]. The regulation of ATP-binding cassette B (ABCB) auxin transporters is also important for proper auxin distribution in the hypocotyl in simulated shade [142].

In the hypocotyls, low R/FR ratio light also controls the localization of PIN3 [140], which plays a key role in tropic responses [143,144]. Analogous to tropic responses, it was hypothesized almost 20 years ago that shade-induced elongation could be produced by a laterally symmetric redistribution of auxin [27,145,146]. In accordance, it has been subsequently demonstrated that a low R/FR ratio light leads to PIN3 lateral localization in the hypocotyl endodermal cells toward the cortical and epidermal cells [140]. 

Interestingly, it has been recently demonstrated that the control of auxin fluxes is essential to coordinate shoot and root growth in response to light cues [141,147]. *PIN1* is expressed at low levels in the hypocotyls of Arabidopsis etiolated seedlings, and it is significantly upregulated upon light exposure, thus suggesting that light may control shoot-to-root polar auxin transport mainly through the regulation of *PIN1* expression in the hypocotyl. Accordingly, it has been shown that *pin1* displays a reduced root length and alterations in the root apical meristem (RAM) that were highly similar to those of plants treated with polar auxin transport inhibitors. Remarkably, the expression of *PIN1* in the hypocotyl is regulated by COP1. Therefore, COP1, whose activity is determined by light, affects shoot-derived auxin levels in the root. This affects root elongation and adapts auxin transport and cell proliferation in the RAM, modulating the intracellular distribution of PIN1 and PIN2 in the root in a COP1-dependent manner [147]. Under simulated shade, a significant downregulation of *PIN1* in the hypocotyl, together with a concomitant reduction in auxin levels in the RAM, has also been observed, indicating that it is likely that a low R/FR light light may activate a PIN1-dependent mechanism, similar to that described in etiolated seedlings [141,147]. Interestingly, it appears that COP1 plays a dual role in the regulation of root growth according to the light present in the environment. Indeed, COP1, on one hand, controls the long-distance transport of auxin, and, on the other hand, regulates local fluxes of auxin in the RAM through different mechanisms [147]. As for the first mechanism, it has been suggested that HY5, which is one of the best characterized targets of COP1, might directly regulate *PIN1* transcription in the hypocotyl [147]. Notably, recent work has shown that HY5 is a shoot-to-root mobile signal involved in the promotion of root growth by light [148,149]. The perception of low R/FR in the shoot also results in a decrease in lateral root (LR) emergence, and it has been proposed that HY5 regulates this process by inhibiting the auxin efflux carrier PIN3 and the influx carrier LIKE-AUX1 3 (LAX3) auxin transporters, which act in concert in the process of LR emergence [149,150].

### 4.3. Auxin Signaling

The Transport Inhibitor Response 1/Auxin Signaling F-Box (TIR1/AFBs) proteins are auxin receptors and are components of the SKP1 CULLIN–FBOX (SCF)-type E3 ligase complex, SCF^TIR1-AFBs^. Auxin binding to SCF^TIR1AFBs^ determines the ubiquitination and degradation of the Auxin/Indole-3-Acetic Acid (Aux/IAA) proteins. Aux/IAAs function as repressors by forming dimers with Auxin Response Factors (ARFs), and their degradation releases the inhibition of ARF transcription factors [151,152]. 

Relevantly, it has been shown that a low R/FR light ratio rapidly and transiently diminishes the frequency of cell division in Arabidopsis leaf primordia through a mechanism that requires TIR1. Consistent with the role of HFR1/SICS1 in the shade avoidance response, the leaf primordium phenotype is enhanced in *hfr1/sics1* mutant seedlings in a low R/FR light ratio (Figure 4). 

The auxin increase perceived through TIR1 results in the upregulation of *Cytokinin Oxidase/Dehydrogenase 6* (*CKX6*), which is a gene encoding an enzyme that catalyzes the irreversible degradation of cytokinin [153,154]. This, in turn, lowers local cytokinin levels, and reduces cell proliferation in developing leaf primordia [133,155]. Further studies are needed to identify the specific ARF(s) that are involved in the induction of *CKX6* by a low R/FR light ratio.

A number of studies have identified auxin-related genes as overrepresented among the genes induced by shade in young seedlings [9,14,23,24,33,49,131,156]. Interestingly, a large fraction of these genes are upregulated in both cotyledons and hypocotyl, thus indicating that shade-induced elongation depends not only on the cotyledon-derived auxin, but also on local hypocotyl signals [131]. Among the auxin-related genes rapidly induced by low R/FR are several early auxin response genes, particularly members of the *Aux/IAA* and the *Small Auxin Up RNA* (*SAUR*) gene families, thus indicating that a number of ARF proteins contribute to the shade avoidance response. Recent work indeed provided evidence that three ARF proteins, ARF6, NPH4/ARF7, and ARF8, together play a key role in the regulation of hypocotyl elongation in a low R/FR environment, as well as in response to other signals, including high temperature [157].

## 5. From Arabidopsis to Crops

The yield of a crop depends to a large degree on its radiation use efficiency and capacity of light interception. At a high planting density, the light interception depends on plant architecture, the degree of mutual shading among plants, and the genetically defined ability of the plant to react to shading, i.e., producing new leaves or reorienting the leaves toward open light [5]. Indeed, several of the effects of the perception of low R/FR signals appear to be negative for yield. Interestingly, despite breeding programs resulting in new cultivars with increased performance under high planting density, many crops still retain the ability to sense and react to canopy shade. For instance, the sensing and reactions to low R/FR, including elongation responses, are present in modern commercial hybrids of maize [158,159,160]. Similarly, the analysis of 10 modern Argentinian wheat cultivars revealed that the selection for yield did not reduced the ability to respond to a low R/FR ratio and diminish the impact of the negative control of productivity [161]. The reduction of these responses may allow increasing plant productivity at a higher density or may provide higher yield at current densities. This could be realized through the selection of natural variants or mutants, as well as by the generation of mutations in critical factor genes by New Breeding Techniques (NBT) or the production of transgenic plants (a.k.a. Genetically Modified Organism, GMO) expressing specific regulators. The latter two approaches require the identification of key regulatory factors. Arabidopsis is an excellent model system to uncover and dissect mechanisms regulating the shade avoidance response, some of which are likely to be conserved during evolution. However, some important differences are emerging from the analysis of other plant species, which have been recently described in several excellent reviews [162,163,164]. It is clear that we have to expand our knowledge of other plant species, especially those representing crop model plants, both for food and energy production. Effective approaches for studying the dynamics of shade avoidance and the identification of critical regulators include genome-wide transcriptional analyses, also taking advantage of the genetic diversity of wild and cultivated species and introgression line (IL) populations produced by their crossing. Here, we briefly review the main results obtained in maize and tomato, which are two economically important mono and dicotyledonous crops, respectively.

### 5.1. Maize

The genome of maize encodes three types of phytochromes (PHYA, PHYB, and PHYC) [165]. PHYB is encoded by two genes (*PHYB1* and *PHYB2*) derived from an ancient tetraploidization event, and both phytochromes contribute differently to distinct physiological aspects of the shade avoidance response [166]. The *phyB1 phyB2* double mutant phenocopies wild-type plants grown in shade, including increased plant height and internode length, reduced tillering, and early flowering [166]. Studies in hybrid maize and teosinte using end-of-day far-red (EOD-FR) light treatments suggested that mesocotyl elongation responses were of the same magnitude [160]. However, a comparison between a modern and an old variety suggested that hybrids that are more productive under high-density plantings may have a reduced auxin response to changes in light quality [159]. The recent data of a genome-wide expression analysis using the maize B73 elite inbred line support this hypothesis [93]. Interestingly, light conditions mimicking canopy shade identical to those utilized by Ruberti et al. to study the process in Arabidopsis [9] were used for the analysis of the shade avoidance response in maize [93]. Consistently, under this light condition, maize seedlings showed an elongated phenotype that was typical of the shade avoidance response. Thereby, the authors were able to compare the dynamics of the transcriptional reprogramming in the two plant species. Two major important differences, among several others, came out from this analysis. First of all, the *YUC* genes, which were strongly induced by low R/FR light in Arabidopsis, were not found regulated in maize. Conversely, *TAA1* was slightly upregulated in maize seedlings, whereas it is downregulated to some extent in Arabidopsis. Coherently, the Gene Ontology (GO) analysis revealed the lack of an enrichment in auxin response genes among those induced by low R/FR light. Furthermore, a genome-wide expression analysis in rice also revealed the lack of induction of auxin response genes in the coleoptile when the seedlings were exposed to low R/FR light [167]. Therefore, it seems possible that the auxin response may have a less important role in monocots, or be a peculiarity of the shade avoidance response in dicotyledonous plants, as confirmed by the large amount of data collected [121,126,168]. A confirmation of such a hypothesis will require a more systematic analysis of monocotyledonous plant species and their undomesticated ancestors, including teosinte. In addition, the comparison of maize and Arabidopsis transcriptional responses also revealed very little overlap between the early response genes, even though hundreds of genes are regulated by low R/FR [93]. In particular, only 20 upregulated and 11 downregulated maize genes have orthologous genes similarly regulated by shade in Arabidopsis. In addition, 19 orthologous gene pairs displayed opposite regulation in response to low R/FR light. Among the upregulated orthologous pairs, there are *ATHB2* and *Gigantea* (*GI*). GI has been implicated in the induction of shade-mediated rapid flowering in low R/FR [169]. The role of ATHB2 in the shade avoidance response has been discussed earlier in this review, and, it is of interest that it is induced by low R/FR light in other plant species [92,167,170,171]. The Arabidopsis *ATHB2* gene is a direct target of the PIF proteins [25,172], and the maize genome encodes for homologs of the Arabidopsis PIF proteins. The constitutive expression of either ZmPIF4 or ZmPIF5 partially rescues the reduced hypocotyl phenotype of the quadruple *pif1 pif3 pif4 pif5* (*pifq*) Arabidopsis mutant, and the overexpression of *ZmPIF5* in Arabidopsis exhibited a constitutive shade avoidance phenotype [173]. Further studies should clarify if the ZmPIFs have any role in the shade avoidance response, including the upregulation of *ATHB2*-like maize genes.

### 5.2. Tomato

Physiological and molecular studies have begun to dissect the effects of neighbor detection and shade avoidance in tomato [92,171,174,175,176]. As other plant species, tomato plants exposed to low R/FR elongate both internodes and petioles more. Unlike other species, tomato plants increase the size of the SAM and incipient leaf primordia, and of the leaf blade when exposed to shade. The alteration of leaf morphology has been observed both in cultivated [129] and wild species [177]. Molecular studies have begun to highlight specific patterns of gene expression in the leaf and stem. Particularly significant is the differential regulation of genes involved in photosynthesis in the leaf and stem, being upregulated and downregulated, respectively [170]. As in the case of maize, the domestication of tomato results in plants that exhibit a reduced shade avoidance response compared to wild tomato species. By means of the introgression analysis of a population arising from a cross between the cultivated tomato M82 and the wild relative *Solanum pennellii*, several loci have been found to affect the strength of shade avoidance, either positively or negatively. The expression analysis of the introgressed lines (ILs) confirmed and extended the molecular data obtained by Casal et al. [170]. In particular, this analysis identified a group of auxin-related genes whose expression correlates with the strength of the shade avoidance response, being upregulated in strong responding and downregulated in tolerant lines, respectively [174]. However, prolonged exposure to shade, while still producing shade avoidance responses, results in normal levels of auxin both in the leaf and stem, although auxin-responsive genes are found upregulated [168]. Similar results are also found in Arabidopsis and soybean [124,178,179], indicating that part of the responses to prolonged exposure to shade is produced by an increased sensitivity to auxin [179]. The analysis of ILs also revealed a very limited number of transcription factor genes regulated by shade; among these genes, only three homologs of *ATHB2* and the homolog of *Ethylene and Salt Inducible 3* (*ESE3*) [174] are induced by shade in Arabidopsis, whereas ESE3 is not regulated in maize [93]. Expression profiling studies in the first emerging leaf primordium exposed to shade light for 28 h also revealed a significant upregulation in the expression of the tomato ortholog of *Shootmeristemless* and other *KNOX*-related genes that are known to promote indeterminacy, and the downregulation of genes involved in leaf differentiation [92].

## 6. Conclusions

Dose-dependent responses to transient and/or persistent stimuli are very common in nature. Generally, a transient behavior with very steep initial upregulation and a subsequent decay region is observed. The overall shape of the response depends on the magnitude of the stimulus received, i.e., it shows a dose-dependent behavior, likely as the product of negative feedback(s). The persistence or the extinction of the response depends on the permanence of the stimulus.

Recent data in Arabidopsis and tomato strongly suggest that the strength of the shade avoidance response depends on auxin. Studies at the molecular level that were conducted mainly in Arabidopsis have highlighted two distinct molecular programs operating in the shade avoidance response. The first one, which is defined as neighbor detection, is characterized by a strong induction of auxin biosynthesis, its accumulation and transport, and transduction of the auxin signal, together with the upregulation of several transcription factor genes and the expression of multiple hormone pathways with distinct and/or overlapping programs taking place in different organs [131]. This molecular response is rapid and transient; it is a “warning signal” that is comparable to a defense response, with the auxin biosynthesis quickly turned off by the intensity of the light reaching the plant, which affects the stability of the negative regulator HRF1/SICS1 [123]. The second program (canopy shade) takes place later on, in part overlaps with the first one, and persists even when the plant is unable to escape shade by the need of the plant to acclimate to the new environmental conditions characterized by a less efficient photosynthetic light. It has been proposed that auxin signaling is also involved in the regulation of this program, likely by a change in the sensitivity to auxin rather than an increase in the concentration of this hormone [25,122,123,124,178,179]. However, intriguingly, the data accumulating in monocotyledonous plant species seem to indicate a reduced or even the lack of an auxin response(s), in spite of the presence of a characteristic shade avoidance response [93,159,167].

It is worth reminding that neighbor detection and canopy shade are both under the strict control of the phytochrome systems through the PIF proteins, and that the whole processes are rapidly reversed by high R/FR light, eventually just by increased irradiance and/or the altered spectral composition of sunflecks perceived through the canopy [156]. Consistently, *ATHB2*, being a direct target of PIF proteins, is rapidly and reversibly regulated by changes in the R/FR light ratio [29], and it is fully induced even by local irradiation [180]. Evidence is accumulating that ATHB2 and its homologs are key regulators of the shade avoidance response, at least in Arabidopsis. Indeed, the overexpression of different members of the HD-Zip II family phenocopies the effect of shade light on distinct organs and flowering, even when the plants are grown in high R/FR [26,27,35,79,86,90]. On the contrary, single and double loss-of-function *HD-ZIP II* mutants display altered growth responses to shade both in the hypocotyl and in the leaf [28,90,94]. In agreement, the expression of a dominant-negative *athb2* mutation in transgenic Arabidopsis and tomato plants results in phenotypic alterations that are suggestive of an overall attenuation of the shade avoidance response [181]. Unfortunately, multiple loss-of-function *HD-Zip II* mutants are difficult to test in shade, since they are strongly altered in embryo, SAM activity, leaf polarity, and gynoecium and fruit development under simulated sunlight conditions [83,98,100], implying a fundamental role of these proteins in the regulation of plant growth and development. Indeed, there are evidences that the alteration of selected HD-Zip II proteins affects at least a regulatory circuit between HD-Zip II and HD-Zip III transcription factors [73,79,83,98,118] and hormones’ signal transduction pathways [101,182]. In addition, evidence exists that a PIF/HD-Zip II genetic module was recruited to carpel development in Arabidopsis [99].

In evolutionary terms, the shade avoidance response appears to be a relatively recent invention that is predominantly found in angiosperms, and it has been considered one of the factors that has contributed to their success [13].

Although the transcriptional program(s) that regulate the developmental responses to shade may be different in distant evolutionary species, it is relevant to emphasize that *ATHB2* and its homologs are the only transcription factor genes regulated by low R/FR light in all of the species that have been analyzed up to today, including poplar [183].

Further work is needed to establish whether ATHB2 and ATHB2-like proteins, together with the PIF proteins, may be considered as the “core regulatory module” recruited to escape and/or adapt to canopy shade.

## Figures and Tables

**Figure 1 plants-07-00102-f001:**
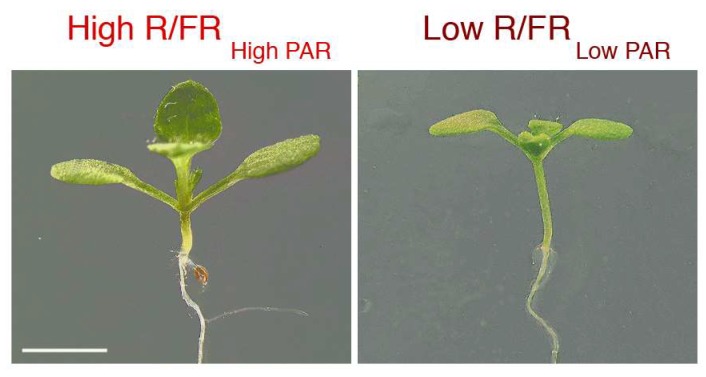
Shade avoidance phenotypes in Arabidopsis seedlings. Seedlings were grown for four days in high red (R)/far-red (FR) _High PAR_ and then either maintained in the same light regime or transferred to low R/FR _Low PAR_ for six days in a 16-h light/8-h dark photoperiod to simulate, respectively, sunlight and shade. Light outputs were as previously reported [9]. Scale bar, 2 mm.

**Figure 2 plants-07-00102-f002:**
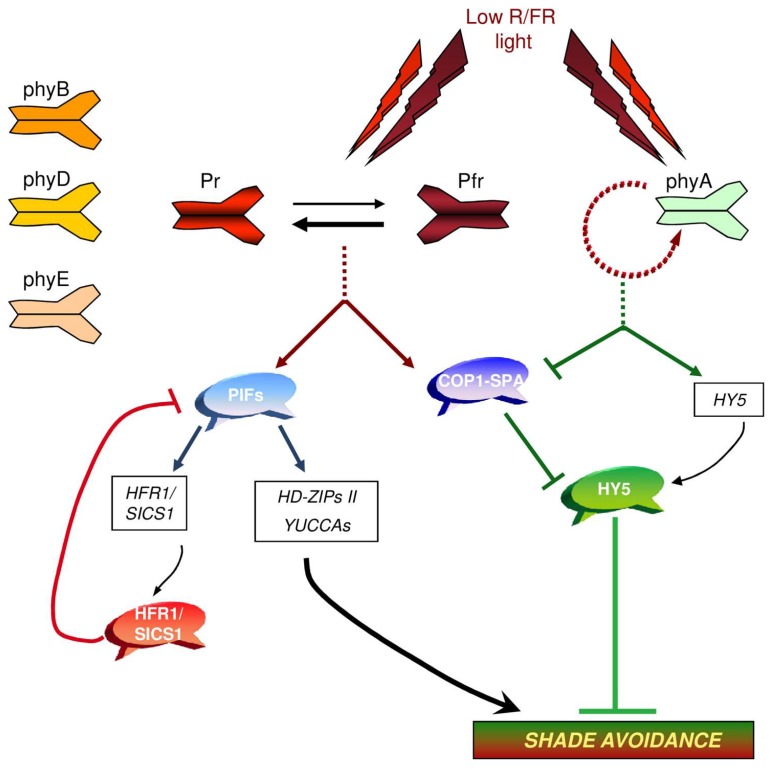
Regulatory routes in the shade avoidance response. Changes in R/FR light causing a shift in the equilibrium between Pr and the FR light-absorbing photo-convertible isoform (Pfr) toward the R light-absorbing photo-convertible isoform (Pr) result in the deactivation of phyB, phyD, and phyE. This, in turn, results in the enhanced stability and/or activity of several phytochrome-interacting transcription factors (PIFs). PIFs, within a few minutes, activate the transcription of *HD-Zips II*, *YUCs*, and *HFR1/SICS1* genes, encoding positive and negative regulators of shade avoidance, respectively. HFR1/SICS1 form non-functional heterodimers with PIF proteins, thereby inhibiting their activity. Shade avoidance is counteracted by the action of phyA, which positively regulates *HY5*, a central regulator of seedling photomorphogenesis. phyA and phyB oppositely affect the activity of COP1/SPA complexes.

**Figure 3 plants-07-00102-f003:**
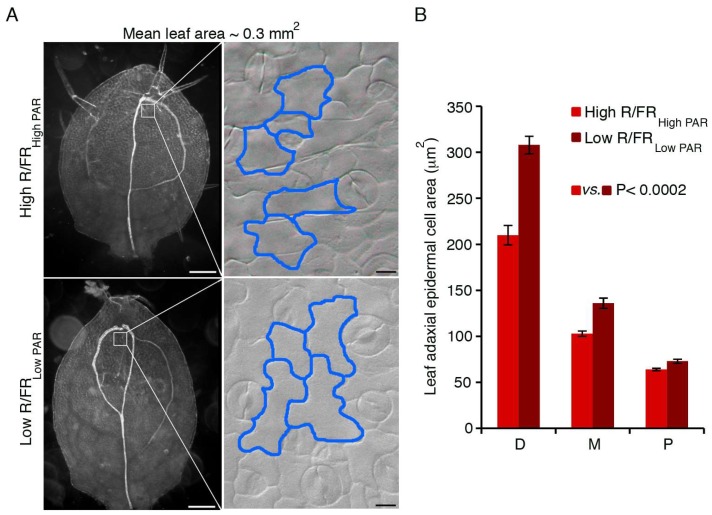
Shade affects adaxial epidermal cell expansion in the Arabidopsis leaf. (**A**) Dark-field images of cleared first/second leaves of wild type grown for eight days in high R/FR _High PAR_ (high R/FR _High PAR_), or for four days in high R/FR _High PAR_ and then for 5.5 days in low R/FR _Low PAR_ (low R/FR _Low PAR_), respectively. The insets show a paradermal view of leaf adaxial epidermis; the borders of a few cells have been highlighted manually with a blue line. Light outputs were as previously reported [9]. Scale bars: (**A**), 100 μm; insets, 10 μm. (**B**) The graph shows the mean epidermal cell area at three positions along the proximo-distal leaf axis, distal (D), median (M) and proximal (P) in the two light conditions. At least 100 adaxial epidermal cells in 10 leaves were analyzed for each condition. Statistical analysis was performed as described [94].

**Figure 4 plants-07-00102-f004:**
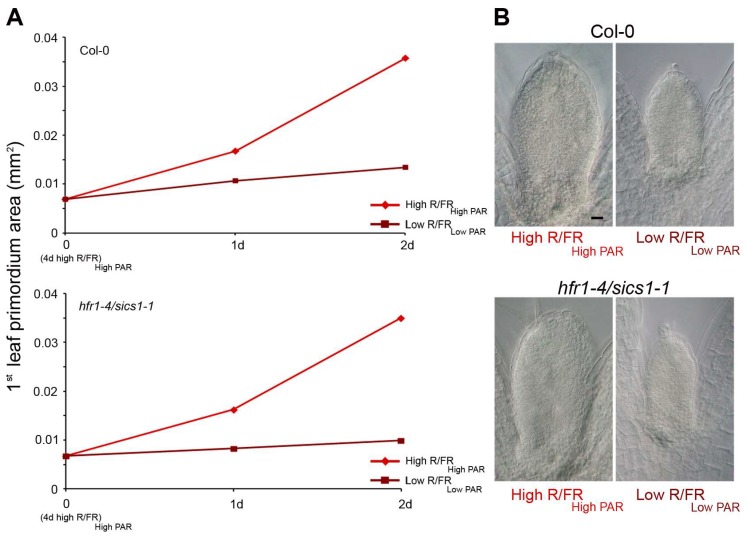
***hfr1/sics1* mutation causes an exaggerated leaf primordium phenotype in shade.** (**A**) *hfr1/sics1* and control (Col-0) seedlings were grown for four days in high R/FR _High PAR_, and then either maintained in the same light regime (red lines) or transferred to low R/FR _Low PAR_ for different times (garnet red lines). The mean area of the first/second leaf primordium was calculated by analyzing 50 samples in each condition. (**B**) Leaf primordia, observed under Differential Interference Contrast (DIC) optics, of *hfr1/sics1* and Col-0 grown for four days in high R/FR _High PAR_, and then either maintained in the same light regime or transferred for two days to low R/FR _Low PAR_. Light outputs were as previously reported [9]. Scale bar, 10 μm.

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
