# Peer review of "Multiple Pathways in the Control of the Shade Avoidance Response"

_plants, 2018, doi:10.3390/plants7040102_

Reviewer 1 Report

The manuscript by Sessa et al., is a comprehensive review of the shade avoidance response in Arabidopsis and crop plants. The paper gives a broad overview of the different photorecptors involved in detected shade followed by an overview of auxin production and signaling. The paper finished by discussing the role of the different families of HD-ZIP transcription factors involved. The manuscript is very well written and a pleasure to read. The individual chapters give a good overview. I only have very few comments:

- line 32 replace distinct with divided

- Line 35 replace "However, among them" with "Among these cues,"

- Line 81 exchange "provoke" with "cause" 

- The chapter on auxin Homeostasis is very PIF-centered. It has been shown that TAA1 and YUC5 are also downregulated by the KANADI1 transcription factor that can repress auxin production (Xie et al., Meta-Analysis of Arabidopsis KANADI1 Direct Target Genes Identifies a Basic Growth-Promoting Module Acting Upstream of Hormonal Signaling Pathways, Plant Phys., 2015). Maybe a sentence on these findings could be added.

- Line 420 change "The yield of a crop somehow depends on its..." to "The yield of a crop depends to a large degree on its...".

Author Response

Reviewer 1

The manuscript by Sessa et al., is a comprehensive review of the shade avoidance response in Arabidopsis and crop plants. The paper gives a broad overview of the different photorecptors involved in detected shade followed by an overview of auxin production and signaling. The paper finished by discussing the role of the different families of HD-ZIP transcription factors involved. The manuscript is very well written and a pleasure to read. The individual chapters give a good overview. I only have very few comments:

-         line 32 replace distinct with divided

Answer

The word "distinct" has been replaced with "divided" (line 32, tracked version);

-         Line 35 replace "However, among them" with "Among these cues,"

Answer

"However, among them" has been replaced with "Among these cues," (lines 34-35, tracked version);

-       Line 81 exchange "provoke" with "cause"

Answer

The sentence "Changes in R/FR light are sensed by phyB, phyD, and phyE and provoke a shift in the equilibrium between Pr and Pfr toward Pr" has been changed with "Changes in R/FR light causing a shift in the equilibrium between Pr and Pfr toward Pr result in a de-activation of phyB, phyD, and phyE" (lines 115-117, tracked version).

-       The chapter on auxin Homeostasis is very PIF-centered. It has been shown that TAA1 and YUC5 are also downregulated by the KANADI1 transcription factor that can repress auxin production (Xie et al., Meta-Analysis of Arabidopsis KANADI1 Direct Target Genes Identifies a Basic Growth-Promoting Module Acting Upstream of Hormonal Signaling Pathways, Plant Phys., 2015). Maybe a sentence on these findings could be added.

Answer

The following sentence has been added in the text:  "Relevantly, TAA1 and YUC5 are directly negatively regulated by KANADI1 (KAN1), a key determinant of abaxial cell fate in the leaf" (lines 991, 992, tracked version).

- Line 420 change "The yield of a crop somehow depends on its..." to "The yield of a crop depends to a large degree on its...".

Answer

"The yield of a crop somehow depends on its..." has been changed to "The yield of a crop depends to a large degree on its..." (line 1164, tracked version).

Reviewer 2 Report

The review article titled “Multiple pathways…..shade avoidance response” attempted to highlight various networks involved in eliciting the shade avoidance responses. In the first few paragraphs, the author summarises the role of Red-Far-red, blue light and UV photoreceptors. The next section indicates how the auxin signalling and transport influences the plant responses to shade. The following sub-subsections are related to various transcription factors involved in the shade avoidance. An additional section on the research involving crops (Maize and tomato) has been presented. The case studies of crops have been referred majorly on two crops- maize and tomato. Overall, the review paper seems appealing because of a) adequate description of shade avoidance responses and b) broader coverage on the references. 

I would suggest the authors move the section on the Auxin to the latter part of the manuscript so that photoreceptors and their downstream transcription factors (HD-ZIPs) signalling would fit together. Moreover, the conclusion section is gearing to auxin pathways.

Author Response

Reviewer 2

The review article titled “Multiple pathways…..shade avoidance response” attempted to highlight various networks involved in eliciting the shade avoidance responses. In the first few paragraphs, the author summarises the role of Red-Far-red, blue light and UV photoreceptors. The next section indicates how the auxin signalling and transport influences the plant responses to shade. The following sub-subsections are related to various transcription factors involved in the shade avoidance. An additional section on the research involving crops (Maize and tomato) has been presented. The case studies of crops have been referred majorly on two crops- maize and tomato. Overall, the review paper seems appealing because of a) adequate description of shade avoidance responses and b) broader coverage on the references. 

I would suggest the authors move the section on the Auxin to the latter part of the manuscript so that photoreceptors and their downstream transcription factors (HD-ZIPs) signalling would fit together. Moreover, the conclusion section is gearing to auxin pathways.

Answer

The order of the sections Auxin and HD-Zips has been changed, as suggested by the Reviewer.